# Multi-Stimulus Responsive Multilayer Coating for Treatment of Device-Associated Infections

**DOI:** 10.3390/jfb13010024

**Published:** 2022-02-28

**Authors:** Wenlong Li, Guanping Hua, Jingfeng Cai, Yaming Zhou, Xi Zhou, Miao Wang, Xiumin Wang, Baoqing Fu, Lei Ren

**Affiliations:** 1Higher Educational Key Laboratory for Biomedical Engineering of Fujian Province, Research Center of Biomedical Engineering of Xiamen, Department of Biomaterials, College of Materials, Xiamen University, 422 Siming Nan Road, Xiamen 361005, China; 20720190153834@stu.xmu.edu.cn (W.L.); hgpailber@163.com (G.H.); 20720201150033@stu.xmu.edu.cn (J.C.); blue_zzyymm@163.com (Y.Z.); xizhou@xmu.edu.cn (X.Z.); 2School of Pharmaceutical Sciences, Xiamen University, Xiamen 361102, China; wangxm@xmu.edu.cn; 3Department of Laboratory Medicine, Xiang’an Hospital of Xiamen University, School of Medicine, Xiamen University, Xiamen 361102, China; 4State Key Laboratory of Physical Chemistry of Solid Surfaces, College of Chemistry and Chemical Engineering, Xiamen University, Xiamen 361005, China

**Keywords:** device-associated infections, multi-stimulus responsiveness, antibacterial coating, wound healing

## Abstract

Antibacterial coating with antibiotics is highly effective in avoiding device-associated infections (DAIs) which is an unsolved healthcare problem that causes significant morbidity and mortality rates. However, bacterial drug resistance caused by uncontrolled release of antibiotics seriously restricts clinical efficacy of antibacterial coating. Hence, a local and controlled-release system which can release antibiotics in response to bacterial infected signals is necessary in antibacterial coating. Herein, a multi-stimulus responsive multilayer antibacterial coating was prepared through layer-by-layer (LbL) self-assembly of montmorillonite (MMT), chlorhexidine acetate (CHA) and Poly(protocatechuic acid-polyethylene glycol 1000-bis(phenylboronic acid carbamoyl) cystamine) (PPPB). The coating can be covered on various substrates such as cellulose acetate membrane, polyacrylonitrile membrane, polyvinyl chloride membrane, and polyurethane membrane, proving it is a versatile coating. Under the stimulation of acids, glucose or dithiothreitol, this coating was able to achieve controlled release of CHA and kill more than 99% of Staphylococcus aureus and Escherichia coli (4 × 10^8^ CFU/mL) within 4 h. In the mouse infection model, CHA releasing of the coating was triggered by infected microenvironment to completely kill bacteria, achieving wounds healing within 14 days.

## 1. Introduction

Nosocomial infection is acquired in healthcare facilities [1]. More than 50% nosocomial infections are related to medical devices [2]. Even with aseptic techniques, microbial communities from the patient’s surface or external environment can be attached to medical devices leading to device-associated infections (DAIs) which is a major health care problem that has not been solved [3]. For instance, the substantial mortality caused by catheter-associated infection have reached 25% [4,5]. A study from the World Health Organization demonstrated that patients in the intensive care unit (ICU) are at a significantly higher risk of acquiring DAIs. About 30% of patients have experienced at least one DAIs with significant associated morbidity and mortality [6].

To prevent the occurrence of DAIs, a promising strategy is to modify the antibiotics on the devices surface [7]. Chlorhexidine acetate (CHA), a kind of common broad-spectrum antibiotic, carries a large number of positive groups that can bind to the anionic sites on the bacteria membrane [8]. This binding can disrupt bacteria membrane potential and cause varying degrees of membrane damage [9]. Therefore, it is difficult for bacteria to develop resistance to CHA [10]. Given the above advantages, CHA is often used as an antibacterial component of the antibacterial coating. For instance, Usha et al. modified CHA on the surface of fabrics and achieved the destruction of bacteria (almost 100%) within 24 h [11]. However, these kinds of antibacterial coatings have obvious defects. While killing bacteria, abundant surface positive charges will cause a large number of bacterial fragments to accumulate, and the antibacterial ability of the coatings will soon fail [12]. At the same time, excessive release of antibiotics also contributes to the development of bacterial resistance [13]. In order to solve the above problems, nano-lamellar MMT, as a material with large specific surface area, good absorbability, good biocompatibility, and drug carrying capability, was used to load CHA to complete slow drug release [14]. MMT can load positively charged substances such as CHA between the lamellar layers through cation exchange effect [15]. However, this drug delivery mode can only play a role of slow release, but cannot achieve the effect of on-demand release and sustained maintenance of antibacterial properties. Therefore, how to maintain the antibacterial ability of cationic antibacterial coating on the medical device surface and significantly reduce the toxicity and bacterial resistance is a major bottleneck in current research [16].

In recent years, stimulus-responsive antibacterial coatings have attracted great attention because of the reduction of the excessive release of antibiotics. This antibacterial coating can be used for topical drug delivery by responding to the bacterial infection microenvironment (e.g., weak acid environment (pH < 5), high peptidoglycan content, and high glutathione content) [17], which will inhibit the development of antibiotics resistance and achieve the long-term antibacterial effect. Most of the previous work on stimuli-responsive antibacterial coatings has focused on responding to a single stimulus (e.g., pH, electric field, light, temperature or bioactive molecules [18,19,20,21,22]). Although such single stimulus-responsive antibacterial coatings have been researched deeply in the treatment of DAIs, it is clear that antibacterial coatings with multi-stimulus responsive ability would be more advantageous and widely applied. Because multiple stimulus occurs simultaneously in a real biological and physical environment [7]. For example, Yang Zhou et al. [23] constructed a surface coating with a multi-stimulus response on a silicon substrate containing thermally responsive components poly(*N*-isopropylacrylamide) (PNIPAAm) and phenylboric acid. The coating can respond to changes of pH, sugar and temperature in the biological environment to release drugs and kill bacteria. Therefore, constructing multi-stimulus-responsive antibacterial coatings that are sensitive to the bacterial infection environment may serve as a good strategy for the treatment of DAIs.

However, most coatings are highly selective to the substrate and are limited to the variety of substrate materials [24]. Therefore, it is necessary to develop a kind of coating that can modify the surface of various substrate. In this process, we need to integrate antibacterial agents and responsive drug delivery systems into the coating material. However, it is not easy to load antibiotics into releasing systems because the weak binding between the matrix and small molecules [25], which would result in bacterial antibiotic resistance [26]. Layer-by-layer (LbL) technology provides a solution to this problem. LbL technology originally refers to a technique of forming self-assembled multilayers on charged substrates by alternating deposition of polyelectrolytes, which was first studied and proposed by R. K. Iler [27]. Since the 1990s, LbL technology has developed rapidly. LbL technology can be widely used in various forms (e.g., spraying or immersion) for surface modification of various materials (e.g., planar or particulate substrates) [28]. It has been proved that LbL assembly could provide a controlled carrier system in respect of releasing antibiotics [29]. LbL self-assembly technology has the ability to fully absorb and retain the biological activity of the drug, which is used to construct a good responsive drug releasing mechanism on the surface of the medical devices.

In this paper, a multi-stimulus responsive multilayer antibacterial coating (MMT-PPPB-CHA)_n_ was prepared by the electrostatic adsorption between the inorganic nano-lamellar MMT, cationic antibacterial agent CHA and PPPB (Figure 1). We studied the drug loading capacity, drug release capacity, and antibacterial activity of (MMT-PPPB-CHA)_n_ coating to prove that the coating can be a versatile material used to modify a variety of substrate, and the coating has good drug loading ability and responsive drug release ability. In addition, a mouse epidermal infection model was established to study the in vivo antibacterial activity of (MMT-PPPB-CHA)_n_ coating. This coating can respond to the microenvironment of bacterial infection (such as weak acid environment (pH < 5), high peptidoglycan content, and high glutathione content) to release cationic CHA, providing a new strategy for solving DAIs.

## 2. Materials and Methods

### 2.1. Materials and Instrumentation

Ethanol absolute (AR Grade, ≥99.7%) and chloroform (AR Grade, ≥99.0%) were purchased from Sinopharm Chemical Reagent Shanghai Co., Ltd. (Shanghai, China). Acetic acid (AR Grade, ≥99.5%), phosphate buffer saline (PBS buffer), and dimethyl sulfoxide (DMSO, molecular biology grade, ≥99.7%), 1-(3-Dimethylaminopropyl)-3-ethylcarbodiimide hydro (EDC, ≥95.0%, C600433-0025), glucose (≥99.8%, A100188-0005) were purchased from Sangon Biotechnology Co., Ltd. (Shanghai, China). *E. coli* (ATCC 43888) and *S. aureus* (ATCC BAA-1721) were purchased from China Center of Industrial Culture Collection, CICC. Chlorhexidine acetate (CHA, ≥99.0%, C107054), dithiothreitol (DTT, ≥99.0%, D104859), cysteamine dihydrochloride (CYS, ≥97.0%, C153647), protocatechuic acid (PCA, ≥97.0%, S30117), polyethylene glycol 1000 (PEG1000, AR Grade, 12803702) were purchased from Aladdin (Co. Ltd. Shanghai, China). MMT k-10 (M813515), 3-carboxyphenyl boric acid (CPBA, ≥99.0%, C804442), 4-Dimethylaminopyridine (DMAP, ≥97.0%, D807273), N-Hydroxysuccinimide (NHS, ≥97.0%, N811124) Ethylene imine polymer (PEI, ≥99.0%, E808878), Deuterium oxide (D_2_O, ≥99.9%, D807644), Methanol-d4 (CD_3_OD, ≥99.9%, M812876), DMSO-d6 (≥ 99.9%, D806935) were purchased from Macklin Biochemical Co., Ltd. (Shanghai, China).

### 2.2. Synthesis of PPPB

Firstly, 1.65 g CPBA (10.0 mmol), 1.53 g EDC (8.0 mmol), and 1.40 g NHS (12.2 mmol) were accurately weighed and dissolved in 100 mL PBS buffer (pH 6.86). After incubation for 2 h under full stirring at room temperature, 0.90 g CYS (4.0 mmol) was added, and then the reaction mixture was continuously stirred at room temperature for 12 h. The precipitate produced by the reaction was filtered, washed several times with PBS and ultrapure water, and then centrifuged twice at 6000 rpm. The product was vacuum dried at 60 °C, and finally recrystallized and purified with methanol to obtain white powder BPBAC.

Secondly, 1.54 g PCA and 2.50 g PEG1000 were accurately weighed and dissolved in 20.0 mL DMSO solution. 1.92 g EDC and 305 mg DMAP were added under stirring conditions, and the reaction was carried out at room temperature for 48 h. After the reaction, the precipitate was removed by filtration, and the filtrate was dialyzed by dialysis bag with molecular weight cut-off 1000. The dialysate was firstly dialyzed by DMSO for 12 h, and then by ultrapure water for 36 h. The dialysate was changed every 6 h during dialysis. After dialysis, the product was precipitated into dark yellow solid and dried in vacuum at 30 °C to obtain PCA-PEG1000.

Finally, bis(phenylboronic acid carbamoyl) cystamine (BPBAC) was dissolved in a small amount of methanol solution, then BPBAC methanol solution and PCA-PEG1000 were fully dissolved in 20.0 mL NaOH aqueous solution (pH 10.0), and stirred at 300 rpm for 12 h at room temperature. At the end of the reaction, dialysis bags with molecular weight cut-off 1000 were used in NaOH solution with pH 10.0 for 48 h, and dialysate was changed every 8 h. Finally, the solution was freeze-dried to obtain a dark yellow colloidal substance, namely PPPB.

### 2.3. Synthesis of (MMT-PPPB-CHA)_n_ Coating

Nanosheet MMT was first prepared. 0.8 g MMT was dissolved in 20.0 mL ultrapure water and 20.0 mL methanol, respectively. After sealing and stirring at room temperature for 1 week, 40.0 mg/mL nanosheet MMT aqueous dispersion and 40.0 mg/mL nanosheet MMT methanol dispersion were prepared for standby.

The substrate was immersed in PEI solution (0.5 mg/mL) for 5 min to form a precursor layer. Then, the composite coating (MMT-PPPB-CHA)_n_ was prepared according to the following steps: (1) The substrate was soaked in 1.0 mg/mL MMT dispersion for 30 s and then dried naturally. (2) Natural drying after immersion in 1.0 mg/mL PPPB solution for 30 s. (3) Natural drying after soaking in 2.0 mg/mL CHA solution for 15 s. In this preparation process, the substrate surface is covered with MMT layer, then PPPB layer, and finally CHA layer, forming a layer (MMT-PPPB-CHA) coating. Repeat the above steps “n” times, (MMT-PPPB-CHA)_n_ coating was prepared on the surface of substrates. The substrate surface coatings only modified with MMT and MMT-CHA were prepared by the same method, denoted as (MMT)_n_ and (MMT-CHA)_n_, respectively, as the control groups. In order to ensure the accuracy and objectivity of performance characterization, the prepared coating was washed with pure water and PBS buffer (pH 7.4) for three times before the test, in order to remove the excess CHA, PPPB and MMT that did not form the coating during the preparation process.

### 2.4. Characterization of Polymers PPPB

The FT-IR spectra of BPBAC, PCA-PEG1000, and PPPB were recorded on a Nicolet 6700 FT-IR spectrometer (Thermo Scientific) in the 4000–600 cm^−1^ range. The ^1^H nuclear magnetic resonance (^1^H NMR) (AVANCE Ⅲ 400 MHz, Bruker, Karlsruhe, Germany) of BPBAC, PCA-PEG1000, and PPPB was recorded at 25 °C using CD_3_OD, DMSO-d6, and D_2_O as solvent separately. The Ultraviolet−visible (UV−vis) absorption spectra of PPPB products with the three stimuli were determined by UV spectro-photometer (UV-2550, Shimadzu, Tokyo, Japan).

### 2.5. Characterization of (MMT-PPPB-CHA)_n_ Coating

The morphology of (MMT-PPPB-CHA)_n_ coating was characterized by a field emission scanning electron microscope (SEM, SU-70, Hitachi Nake high-tech enterprise, Tokyo, Japan) at an acceleration voltage of 5 kV. X-ray diffraction (XRD) was performed with the Axs D8-A25 advance (Bruker, Karlsruhe, Germany). The Fourier transform infrared (FTIR) spectra assay was performed using a Nicolet iS10 (Thermo Scientific, Waltham, MA, USA) with the KBr pellet technique. UV−vis absorption spectra were acquired with the UV spectrophotometer (UV-2550, Shimadzu, Karlsruhe, Germany, Japan). MMT, PPPB, and CHA were prepared into 0.5 mg/mL solution. (MMT-CHA)_3_ and (MMT-PPPB-CHA)_3_ coatings were scraped off the substrate and dispersed in water to form a suspension (0.5 mg/mL). The Zeta potential of samples was measured by Malvern Mastersizer 2000 particle size and potential analyzer (Mastersizer2000/MAL1012737, Spectris, Shanghai, China). First of all, the test mode was set as potential test in Zetasizer Software (test temperature: 25 °C, number of repeated tests: 3 times, and potential sample pool model: Dts1070). Then, the solution or suspension was fully shaken and injected at the calibration line of the sample pool. Then, the sample pool was put into the analyzer. Finally, the potential values measured in the three experiments were averaged as the final sample Zeta potential. The water contact angle of samples was measured by a dynamic water contact angle measuring instrument (OCA20, Dataphysics, Germany, Esslingen, Germany).

### 2.6. The Standard Curve of CHA

The concentration of CHA solution prepared was 0.1 mg/mL, 0.05 mg/mL, 0.025 mg/mL, 0.01 mg/mL, 0.005 mg/mL and 0.0025 mg/mL, respectively. The absorbance of CHA at 252 nm was measured by a UV-VIS spectrophotometer (UV-2550, Shimadzu, China) with three times to take the average value. Then the standard curve of CHA concentration and absorbance was drawn.

### 2.7. CHA Loading Capacity of (MMT-PPPB-CHA)_n_ Coating

Five groups of (MMT-PPPB-CHA)_n_ coatings (*n* = 2, 4, 8, 12, and 16) were constructed on the CA films with an area of 2 cm^2^ according to the above steps. 5.0 mL CHA solution with the concentration of 5.0 mg/mL. To facilitate the calculation, 0.5 mL CHA (5.0 mg/mL) was used in preparation. After the preparation, we collected the remaining CHA solution and tested the absorbance of CHA at 252 nm. The relationship between drug loading (µg/cm^2^) and layers number was further calculated. The calculation formula is:$${\mathrm{CHA}\;\mathrm{content}\;\mathrm{per}\;\mathrm{unit}\;\mathrm{area}\;(\mu g/\mathrm{cm}}^{2}\;)=\frac{\mathrm{Total}\;\mathrm{dose}\;(\mu g)\;-\;\mathrm{Remaining}\;\mathrm{dose}\;(\mu g)}{{\mathrm{Total}\;\mathrm{area}\;(\mathrm{cm}}^{2})}$$

### 2.8. Sustained Drug-Releasing Capacity of (MMT-PPPB-CHA)_3_ Coating

The coating (1 × 2 cm^2^) was put into 5.0 mL PBS buffer. 0.5 mL of the release medium was taken out successively at 1, 2, 4, 12, 24, 48, and 72 h, and the same amount of PBS buffer was immediately replaced after taking. The absorbance of CHA was measured at 252 nm by an UV-VIS spectrophotometer (UV-2550, Shimadzu, China), and the CHA release kinetics curves of the different coating were obtained within 72 h.

### 2.9. Multi-Stimulus Responsive Ability of (MMT-PPPB-CHA)_3_ Coating

The (MMT-PPPB-CHA)_3_ coating (1 × 2 cm^2^) was placed in 5.0 mL release media with different pH values (5.5, 6.0, 7.0, and 8.0), different concentrations of glucose solutions (0, 0.5, and 1.0 mg/mL) and different concentrations of DTT solutions (0, 0.5, and 1.0 mg/mL), respectively. 0.5 mL release medium was taken out successively at 1, 2, 4, 12, 24, 48, and 72 h, and an equal amount of release medium with corresponding pH value was immediately replaced after taking. The absorbance of the CHA was measured at 252 nm by an UV-VIS spectrophotometer (UV-2550, Shimadzu, China), and the release kinetics curve of CHA was obtained according to the standard curve.

To further explore the synergistic drug release behavior of (MMT-PPPB-CHA)_3_ coating, orthogonal test L_9_ (3^4^) (three factors and three levels) was used to study the synergistic drug release effect of (MMT-PPPB-CHA)_3_ coating under the different conditions. The orthogonal test results were analyzed as shown in Table 1.

### 2.10. Preparation of Bacteria Suspensions

*E. coli* (ATCC 43888) and *S. aureus* (ATCC BAA-1721) were cultured in 10 mL LB broth at 37 °C for 6 h. After culturing, the bacterial sediments were resuspended and washed with normal saline (0.9%, pH 7.4). Finally, the bacteria suspension was diluted to an optical density of 0.1 at 600 nm (OD600 = 0.1)

### 2.11. In Vitro Antibacterial Activity of (MMT-PPPB-CHA)_3_ Coating

The coated or uncoated substrate was co-cultured with bacteria suspension at different pH values (6.0, 7.0 and 8.0), different concentrations of DTT solutions (0, 0.5, and1.0 mg/mL), and glucose solutions (0, 0.5, and 1.0 mg/mL) for 2 h, respectively. Then, 100 µL of the bacteria suspension was taken out and inoculated on AGAR medium. After 16 h of static culture, the number of colonies was counted and digital photos of culture plates were taken.

Fluorescein isothiocyanate (FITC) was dissolved in DMSO to prepare the 1 mM FITC solution. Live bacteria were labeled with 100 μL FITC solution and incubated with the coated or uncoated substrate at different pH values (6.0, 7.0 and 8.0), different concentrations of DTT solutions (0, 0.5, and 1.0 mg/mL), and glucose solutions (0, 0.5, and 1.0 mg/mL) for 2 h, respectively. The bacteria suspension was dropped into the surface dish specially used for laser confocal microscopy, and the survival of the bacteria was further observed under a laser confocal microscopy (SP8-STED 3X, Leica, German).

The bacteria were fixed with 2% glutaraldehyde aqueous solution, and then the gradient dehydration was carried out with 10%, 20%, 30%, 40%, 50%, 70%, 90%, 100% ethanol aqueous solution successively. The growth and adhesion of bacteria on the surface of the coating material were observed under a SEM (SU-70, Hitachi Nake High-Tech Enterprise, Tokyo, Japan) after dehydration.

### 2.12. Long-Term Antibacterial Performance of (MMT-PPPB-CHA)_3_ Coating

Sterilized CHA, (MMT-CHA)_3_ and (MMT-PPPB-CHA)_3_ coated and uncoated substrates were placed in 5.0 mL bacteria suspension, respectively. Take 100 μL of bacteria solution from each group every 4 h and inoculate them into the AGAR medium for colony counting. At the same time, the bacteria liquid of each group was discarded, cleaned with sterile normal saline, and 5.0 mL of bacteria suspension was added again to continue co-culture. Repeat the preceding steps six times.

### 2.13. In Vivo Antibacterial Activity and Full-Thickness Skin Defect Healing

Female BALB/c mice (8 weeks, 15–20 g) were obtained from Laboratory Animal Center of Xiamen University and divided into five groups with three mice in each group: (1) normal saline (control); (2) uncoated substrate; (3) CHA coated substrate; (4) (MMT-CHA)_3_ coated substrate; (5) (MMT-PPPB-CHA)_3_ coated substrate.

10 μL of *S. aureus* and *E. coli* solution with a concentration of 1.0 × 10^6^ CFU/mL was subcutaneously injected into the back of mice. 48 h later, obvious suppurative infection symptoms could be observed at the bacteria inoculated site on the back of mice. The suppurative infection site was cut open with a sterilized scalpel to form an open resection wound with a diameter of about 6 mm. Corresponding materials were affixed to the wound of each mouse and fixed with medical tape. The materials were removed 24 h later.

All mice were weighed every 24 h, and the wound healing of the subcutaneous infected site was observed and photographed at certain intervals. ImageJ was used to measure the change of wound area at different periods to calculate the wound healing rate [30], which was calculated as follows:$$\mathrm{Wound}\;\mathrm{healing}\;\mathrm{rate}\;\left(\%\right)=(1\;-\frac{\mathrm{wound}\;\mathrm{area}\;\mathrm{on}\;\mathrm{day}\;n}{\mathrm{wound}\;\mathrm{area}\;\mathrm{on}\;\mathrm{day}\;0}\;)\times 100\%$$

Finally, the mice were killed, and the bacterial concentration of the infected tissue was observed on the AGAR plate. This in vivo experiment has been approved by the Institutional Animal Care and Use Committee of Xiamen University. Approval date: 26 February 2018; Approval code: XMULAC20180003.

### 2.14. Statistical Analysis

Data are expressed as mean ± S.D. At least three independent experiments were performed for in vitro experiments. Statistical analysis was carried out using SPSS version 18.0, Origin 2021 and GraphPad Prism 7.00. Data were analyzed by two-sided student t-tests for comparison of two groups and one-way ANOVA for multiple groups.

## 3. Results and Discussion

### 3.1. Synthesis of the Multi-Stimulus Responsive Polymer PPPB

We firstly synthesized a multi-stimulus responsive polymer PPPB based on phenylborate ester bond and disulfide bond (Figure 2a) to construct a multi-stimulus responsive antibacterial coating. PPPB was synthesized through dynamic covalent reactions between BPBAC (Appendix A) and PCA-PEG1000 (Appendix A) under alkaline conditions (pH 10.0). The structure of PPPB was identified based on IR and ^1^H NMR spectrum. The position and peak area ratio (1:1) of the characteristic peaks (δ 2.9 ppm and δ 3.6 ppm) indicated that PPPB possessed the characteristic structure of both BPBAC and PCA-PEG1000 (Figure 2b). Moreover, IR spectrum of PPPB also showed the characteristic peaks of BPBAC and PCA-PEG1000 at 2919 cm^−1^ (C–H) and 1641 cm^−1^ (C=O) (Figure 2c). More importantly, the disappearance of characteristic peaks at 1347 cm^−1^ indicated the phenylboronic acid was transformed to boronate ester, further confirming the successful synthesis of PPPB.

Due to the presence of boronate ester and disulfide bonds, polymer PPPB should exhibit pH-, glucose-, and redox-responsive features theoretically (Appendix A). Therefore, the multi-stimulus responsive degradation behavior of PPPB was explained by UV-absorption, IR and ^1^H NMR spectrum. As shown in Appendix A, PPPB displayed a distinct corresponding UV peak with an absorption maximum at 300 nm. However, the intensity of the peak decreased with the addition of PBS buffer (pH 6.0), glucose aqueous solution (0.5 mg/mL) and DTT aqueous solution (0.5 mg/mL), indicating that the disulfide bond content in the product is reduced. In order to further analyze the stimulus responsiveness of PPPB, we purified the products by dialysis (molecular weight cut-off: 1000 Da), which means that the product obtained by dialysis should be the part containing PEG-1000 fragment. Disulfide bonds in polymer PPPB break when DTT stimulus responsiveness occurs. According to Figure 3, the intercepted part of the broken PPPB should contain part of the structure of BPBAC molecule, which means the existence of the characteristic peak of -S-CH_2_- (δ 2.9 ppm) in the ^1^H-NMR spectrum. On the contrary, the disappearance of the characteristic peak of-S-CH_2_- (δ 2.9 ppm) and the existence of characteristic peak of -CH2-CH2 (δ 3.6 ppm) indicates the intercepted part of the broken PPPB is PCA-PEG1000, which means that pH/redox stimulus responsiveness have occurred.

### 3.2. Construction of the of (MMT-PPPB-CHA)_n_ Coating

Based on multi-stimulus responsive polymer PPPB, we designed a layer-by-layer assembly coating with the specificity of the response to the bacterial infection microenvironment (weak acid environment (pH < 5), high peptidoglycan content, and high glutathione content) (Figure 1). We chose cellulose acetate membrane (CA), polyacrylonitrile membrane (PAN), polyvinyl chloride membrane (PVC), and polyurethane membrane (PU) as substrates to prepare the coating (MMT-PPPB-CHA)_3_ because they are common medical materials in the clinic [31]. According to the color change of the substrate surface, it is preliminarily concluded that (MMT-PPPPB-CHA)_3_ coating could be prepared on various substrate materials. As shown in Appendix A, the surfaces of coated groups appeared even rougher than uncoated groups. Moreover, the change of water contact angle can prove that the coating can obviously improve the hydrophilicity of the substrate, which is helpful to improve the affinity of medical devices to human body (Appendix A). According to the cross-section of the coating (Figure 4a–c), (MMT-PPPB-CHA)_3_ coating could present a thick and loose layer stacking state. The reasons could be explained by zeta potential (Figure 4d). After the coatings have been successfully prepared on the substrate, they are gently scraped off and configured to a 0.5 mg/mL suspension (the solvent is pure water). Zeta potential of the sample was measured by Malvern Mastersizer 2000. Both MMT and PPPB exhibit electronegativity, which was very favorable for loading the positively charged CHA with electrostatic interaction. In addition, (MMT-CHA)_3_ exhibited a weak positive charge after LbL self-assembly of MMT and CHA, which also proved the possibility of PPPB participating in electrostatic interaction. Finally, (MMT-PPPB-CHA)_3_ coatings were negatively charged, confirming the electrostatic interaction between MMT, PPPB, and CHA. This interaction could be beneficial for CHA loading and enlarge the layer spacing of the coating. It can be seen from Figure 3e that the (001) diffraction peak of (MMT)_3_ and (MMT-CHA)_3_ are very obvious, which are corresponded to an interlayer d spacing of 1.2357 nm and 1.3043 nm respectively, according to Bragg’s equation. The (001) interlayer spacing of (MMT)_n_ increased about 0.0686 nm after co-deposition with CHA, indicating that some of CHA was loaded between MMT layers by ion exchange [25,32]. However, the (001) diffraction peak of (MMT-PPPB-CHA)_3_ can not be observed, indicating that the addition of PPPB completely destroyed the layered ordering structure of MMT. Finally, FT-IR absorbance spectrum demonstrated that (MMT-PPPB-CHA)3 coating was prepared successfully (Figure 4f).

### 3.3. CHA Loading Capacity and Releasing Kinetics of (MMT-PPPB-CHA)_n_ Coating

Herein, CHA loading capacity and kinetics of CHA releasing in (MMT-PPPB-CHA)_n_ coating were studied. As shown in Appendix A, the drug loading of (MMT-PPPB-CHA)_n_ coating had a linear relationship with the number of layers (n). This result was in line with our expectation that number of layers could regulate the CHA loading dosages in (MMT-PPPB-CHA)_n_ coating. According to Appendix A, the drug release rate of (MMT-PPPB-CHA)_n_ coating was less than 30%, while the drug release rate of (MMT-CHA)_n_ was more than 60% within 72 h. This indicated that (MMT-PPPB-CHA)_n_ coating possessed good sustained release capacity.

### 3.4. Multistimulus-Responsive Drug Release Behavior of (MMT-PPPB-CHA)_n_ Coating

The ability of sustained release alone does not play a role in inhibiting bacterial resistance. Prolonged exposure to drugs can encourage bacteria to develop drug resistance. Therefore, a controlled drug delivery system in response to bacterial infection microenvironment (weak acid environment (pH < 5), high peptidoglycan content, and high glutathione content) is very important. Considering the presence of multi-stimulus responsive PPPB, (MMT-PPPB-CHA)_3_ coating should theoretically be capable of responsive drug release. Therefore, the multi-stimulus responsive drug release behavior of (MMT-PPPB-CHA)_3_ coating was studied (Figure 5). As shown in Figure 5a, the release rate of CHA was only about 20% in the alkaline or neutral phase (pH 8.0–7.0). When pH reduced to 5.5, the CHA release rate increased to 40% within 72 h. The lower the pH value, the higher the CHA release rate. Moreover, according to Figure 5b,c, the CHA release rate increased with the increased glucose and DTT concentration. When glucose concentration was 1.0 mg/mL, the CHA release rate can reach about 35% and 65% within 72 h respectively. This responsive drug release behavior of the (MMT-PPPB-CHA)_3_ coating was related to the dynamic boronate ester and disulfide bonds in the polymer PPPB. The dynamic covalent bonds in PPPB could be reversibly fractured in weak acid environment (pH < 7.0), high peptidoglycan solution (0.5 mg/mL), and high glutathione solution (0.5 mg/mL). When bond fracture occurs in PPPB, the layered structure of (MMT-PPPB-CHA)_3_ coating gradually became loose and the gradual release of the CHA was started.

The above results showed that (MMT-PPPB-CHA)_3_ coating exhibited responsive drug release ability in different pH (5.5, 6.0, 7.0 and 8.0), concentrations of glucose (0, 0.5, 1.0 mg/mL) and DTT (0, 0.5, 1.0 mg/mL) solutions, respectively. However, these factors exist together in a real infectious environment. Therefore, we studied the influence of these three factors (pH, glucose, and DTT) on CHA release rate. Based on the investigation of single factors, the three factors, namely pH (factor A), glucose (factor B), and DTT (factor C) were selected. CHA release rate was regarded as investigating indicator for screening the optimal combination of three factors through the orthogonal test L_9_ (3^4^) (three factors and three levels) as shown in Table 1 [33]. The average drug release rate was the highest in group c and the lowest in group h, while the drug release levels of groups a, b, d, e, and f were similar. It indicated that the optimal combination of factors to promote CHA release was pH 6.0, glucose 1.0 mg/mL and DTT 1.0 mg/mL. However, the effects of the three factors on the drug release rate of the coating also have a certain order. To explore this order, we analyzed the extremum values (R) which indicates the importance degrees of three factors in Table [34]. By comparing the R-value, the significance order of the three factors on the average CHA release rate was pH > DTT > glucose. Therefore, it can be concluded that pH had the greatest influence on the drug release of the coating.

### 3.5. In Vitro Antibacterial Activity of (MMT-PPPB-CHA)_n_ Coating

Given the results of orthogonal test, we further studied the antibacterial properties of (MMT-PPPB-CHA)_3_ coating in vitro. As shown in Figure 6a,b, bacterial mortality rate of (MMT-PPPB-CHA)_3_ group reached about 99% in 4 h at a normal environment against *S. aureus* and *E. coli*. When coating degradation triggered by weak acid environment (pH 6.0), glucose solution (1.0 mg/mL), and DTT solution (1.0 mg/mL), (MMT-PPPB-CHA)_3_ coating could kill 99% of *S. aureus* and *E. coli* only in 2 h. We speculated that the reason for this might be the high CHA releasing rate of the (MMT-PPPB-CHA)_3_ coating. According to Appendix A, the minimum release amount of CHA was 23 μg (*n* = 3). This dose (23 μg/mL) was still above the MIC of CHA (1.0 μg/mL) according to the literature we reviewed [35]. This caused the bacteria to be killed more quickly in higher concentrations of CHA.

The efficient bactericidal ability can inhibit the development of bacterial resistance, but it does not mean that the medical device surface can be better protected. In particular, cationic antimicrobial modified surfaces are more likely to adsorb bacterial debris. If the dead bacteria can’t be cleaned up quickly, they can provide an attachment site for other living bacteria. In this case, the antibacterial surface can quickly fail and the medical device becomes contaminated. Previous results have proved that the layered self-assembly (MMT-PPPB-CHA)_3_ coating has strong electronegativity and hydrophilicity. According to the literature, these properties should endow the coating with good anti-adhesion capacity [18]. Herein, bacterial adhesion on (MMT-PPPB-CHA)_3_ coating was observed in PBS buffer (pH 5.5, 6.0, 7.0 and 8.0), glucose aqueous solution (0.5 and 1.0 mg/mL) and DTT aqueous solution (0.5 and 1.0 mg/mL). FITC labelled *S. aureus* (2 × 10^6^ CFU/mL) and *E. coli* (2 × 10^6^ CFU/mL) were co-cultured with the (MMT-PPPB-CHA)_3_ coating. As shown in Figure 7a, Appendix A, (MMT-PPPB-CHA)_3_ coating could effectively inhibit the growth of bacteria within 2 h in different solutions (PBS buffer (pH 5.5, 6.0, 7.0 and 8.0), glucose aqueous solution (0.5 and 1.0 mg/mL) and DTT aqueous solution (0.5 and 1.0 mg/mL)), respectively. The number of visible bacteria was significantly reduced (99% reduction) (Appendix A). Moreover, only a few bacteria fragments could be on the (MMT-PPPB-CHA)_3_ coating (Figure 7b, Appendix A). By contrast, the bacteria survived well on the uncoated substrates (Figure 7c, Appendix A).

After proving the anti-adhesion property, we further explored the recyclability of the (MMT-PPPB-CHA)_3_ coating. The substrate covered with different coatings was reused for six cycles (4 h as 1 cycle) (Figure 8). The concentration of the bacteria increased to 10^8^–10^9^ CFU/mL after 4 h in the uncoated group. For CHA coating (Figure 8b), the complete killing of *S. aureus* and *E. coli* was achieved in the first cycle and lost its antibacterial properties in the following cycle. As shown in Figure 8c, the (MMT-CHA)_3_ coating achieved effect bacterial killing in two cycles but began to lose its antibacterial ability in the third cycle. According to Figure 8d, (MMT-PPPB-CHA)_3_ coating completely killed *S. aureus* and *E. coli* in four cycles. In the fifth and sixth cycles, low concentrations (10^3^ CFU/mL) of *S. aureus* and *E. coli* appeared on the surface of the coating, respectively. It could be thus concluded that (MMT-PPPB-CHA)_3_ coating can maintain a long-term antibacterial effect.

### 3.6. In Vivo Antibacterial Activity and Full-Thickness Skin Defect Healing of (MMT-PPPB CHA)_n_ Coating

Given the highly effective antibacterial efficacy of (MMT-PPPB-CHA)_3_ coating in vitro, we studied its in vivo antibacterial performance in an infected mouse model. The process of animal experimentation was depicted in Figure 9a. We inoculated the back of the mouse with bacteria, and cut it open with a scalpel to form an open resection of the infected wound where there were obvious signs of suppurative infection. Then, different coatings were used for treatment. The wound healing of the subcutaneous infected site in mice was observed and photographed on days 1, 5, 10, and 14, respectively, as shown in Figure 9b. The wounds treated with CHA, (MMT-CHA)_3_ and (MMT-PPPB-CHA)_3_ coatings showed no further deterioration and a good healing trend. As shown in Figure 9c,d, (MMT-PPPB-CHA)_3_ coating treated wounds healed 91–97% after 14 days, respectively. However, the infected wounds in the untreated group and the blank substrate group were basically in a state of severe infection. Some symptoms of suppurative infection still existed after 14 days. The healing rates of *S. aureus* and *E. coli* infected wounds were 63% and 72%, respectively. After 14 days, the tissues of the infected site were fully soaked in sterile normal saline and inoculated on AGAR plates to observe the bacteria colony growth, as shown in Appendix A. It was observed that there were still many bacteria living in the infected sites of mice in the untreated group and blank group, indicating the infection did not been eliminated. By contrast, the wound tissue treated with CHA, (MMT-CHA)_3_ and (MMT-PPPB-CHA)_n_ coatings was free of any bacteria.

In clinical practice, medical instruments (e.g., peripherally inserted central catheter and urethral catheter) are exposed to wounds on the surface of the body for a long time [36]. During this time, bacteria have a chance to contaminate medical equipment and invade the body to cause infection [36]. Therefore, it is very important for the coating of medical devices to have high antibacterial properties and promote wound healing. Wound healing is a complex and orderly process consisting of four stages: hemostasis, inflammation, proliferation, and wound remodeling. Herein, H & E staining was applied to study the process of wound healing from a histological perspective (Figure 9e). The skin tissues in the untreated group and the blank substrate group exhibited acute inflammatory responses and displayed very little sprouting angiogenesis on the 14 days [37]. The results showed that the wound healing was in a transitional stage between inflammation and regeneration, indicating the infections in the wound were not resolved. Differently, the skin tissues treated (MMT-PPPB-CHA)_3_ coatings had regenerated hair follicles and capillaries. It is worth noting that the tissue treated with (MMT-PPPB-CHA)_3_ coating showed better hair follicle regeneration, while CHA and (MMT-CHA)_n_ group showed more angiogenesis. Considering the wound can regenerate a large number of blood vessels during the regeneration stage, and complete the degeneration of some blood vessels and generate hair follicles during the remodeling stage. The presence of more hair follicles indicated the best wound recovery in (MMT-PPPB-CHA)_3_ group. In conclusion, a comparison of wound healing of each group showed that (MMT-PPPB-CHA)_3_ coating group healed best no matter *S. aureus* or *E. coli* infected wounds, indicating that (MMT-PPPB-CHA)_n_ coating had good biocompatibility (Appendix A). Moreover, it had the best bactericidal effect on the subcutaneous infected site, thus accelerating the elimination of infection and promoting wound repair.

## 4. Conclusions

In this study, a multi-stimulus responsive multilayer coating with controlled release of CHA was constructed for the treatment of DAIs. The in vitro studies highlighted the pH-, glucose- and DTT-responsive abilities of (MMT-PPPB-CHA)_n_ coating, and its antibacterial capacity. In medium with pH 6.0, glucose 1.0 mg/mL and DTT 1.0 mg/mL, the coating showed extremely high bactericidal ability (99% reduction). In real clinical scenario, the coating is intended to address bacterial infections that occur when medical devices (such as urinary catheters or peripheral venous catheters) are connected to open wounds in humans. The three components involved in the coating are biocompatible components. In vivo studies also show that the coating can kill bacteria quickly and promote wound healing within 14 days, which supports the coating as a non-toxic and highly bactericidal material. In addition, (MMT-PPPB-CHA)_n_ coating could be reused 6 times with antibacterial ability, reducing the pain caused by frequent dressing changes for patients. However, it needs further research that whether the coating still has good antibacterial and biocompatibility in case applied for the stents and prosthetic devices in vivo.

## Figures and Tables

**Figure 1 jfb-13-00024-f001:**
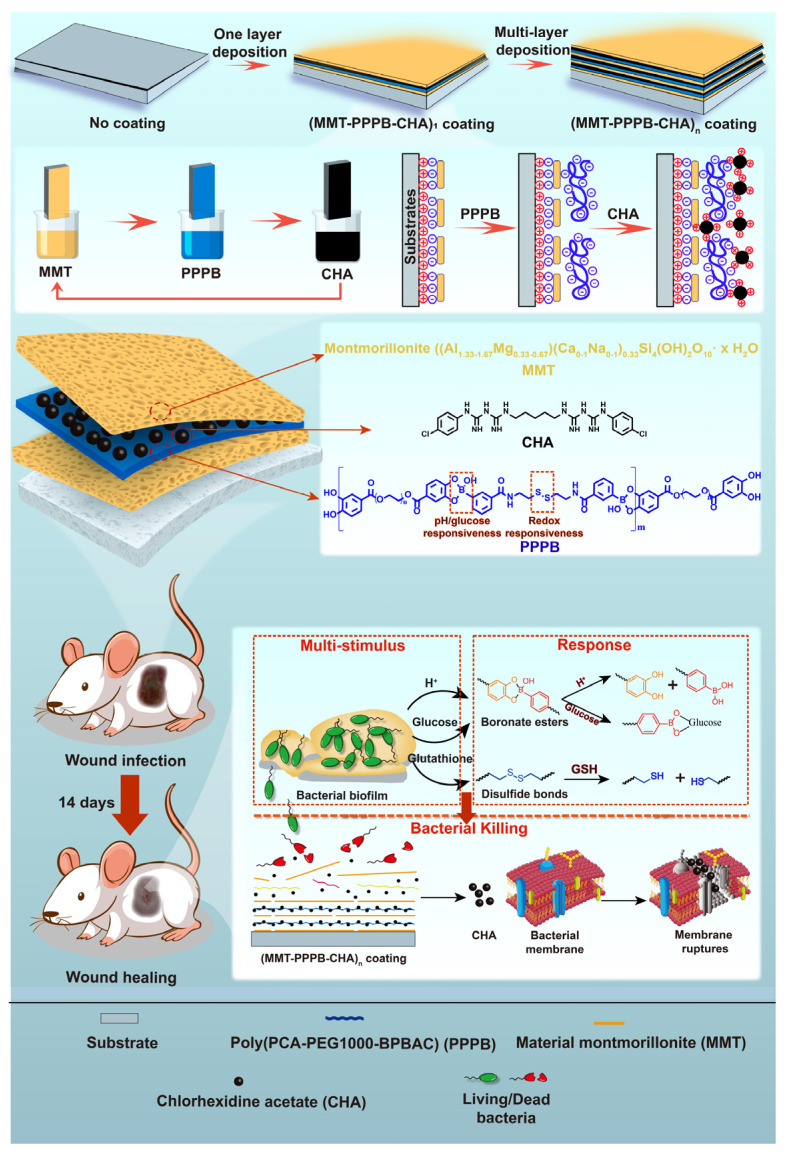
Schematic illustration of the multi-stimulus responsive multilayer antibacterial coating (MMT-PPPB-CHA)_n_.

**Figure 2 jfb-13-00024-f002:**
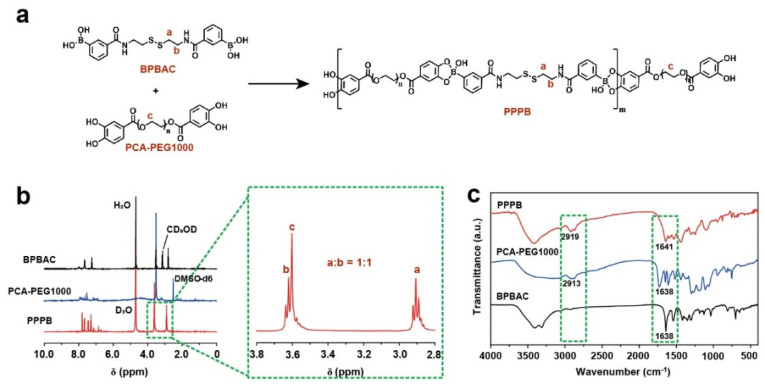
(**a**) Chemical structural formula of polymer PPPB; (**b**) ^1^H-NMR spectrums of PPPB, PCA-PEG1000 and BPBAC; (**c**) FT-IR absorbance spectrums of PPPB, PCA-PEG1000, and BPBAC.

**Figure 3 jfb-13-00024-f003:**
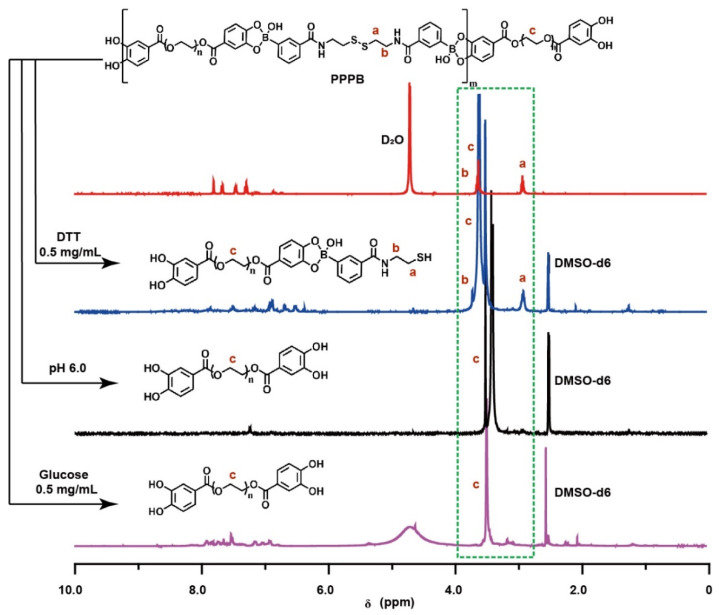
^1^H-NMR spectrums of PPPB in PBS buffer (pH 6.0), glucose aqueous solution (0.5 mg/mL) and DTT aqueous solution (0.5 mg/mL).

**Figure 4 jfb-13-00024-f004:**
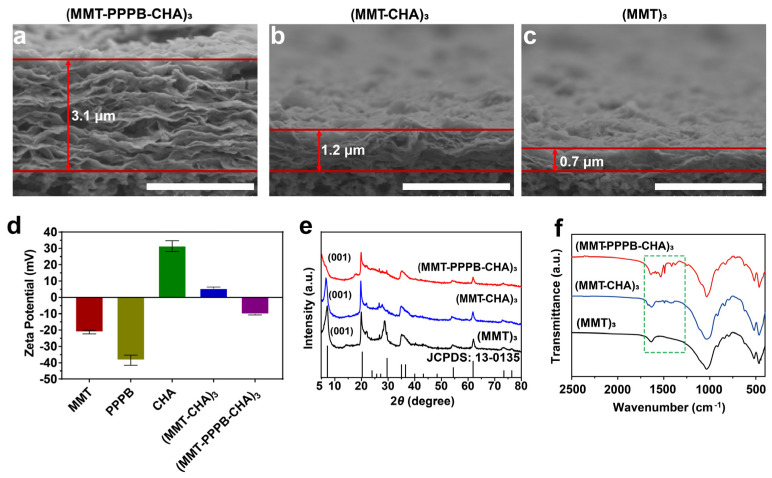
The SEM micrograph of cross-sectional of (**a**) MMT-PPPB-CHA)_3_, (**b**) (MMT-CHA)_3_, and (**c**) (MMT)_3_ coating. Scale bar: 3 μm; (**d**) Zeta potential of MMT, PPPB, CHA, (MMT-CHA)_3_ and (MMT-PPPB-CHA)_3_; (**e**) XRD patterns and (**f**) FT-IR absorbance spectrums of (MMT)_3_, (MMT-CHA)_3_, (MMT-PPPB-CHA)_3_.

**Figure 5 jfb-13-00024-f005:**
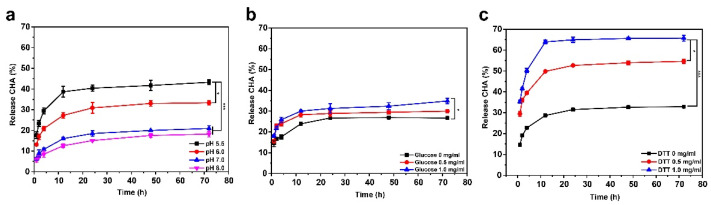
CHA release curves of (MMT-PPPB-CHA)_3_ within 72 h in (**a**) different pH (5.5, 6.0, 7.0 and 8.0), (**b**) concentrations of glucose (0, 0.5, 1.0 mg/mL) and (**c**) DTT (0, 0.5, 1.0 mg/mL) solutions. ** p <* 0.05, **** p* < 0.001.

**Figure 6 jfb-13-00024-f006:**
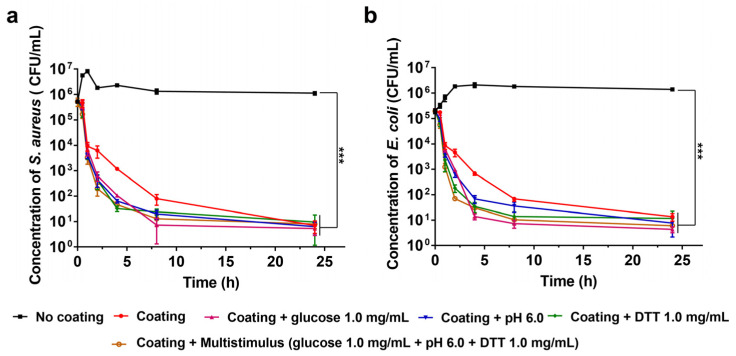
Stimulation-responsive killing effects of (MMT-PPPB-CHA)_3_ on high concentrations of (**a**) *S. aureus* and (**b**) *E. coli* after incubating for 24 h. **** p <* 0.001.

**Figure 7 jfb-13-00024-f007:**
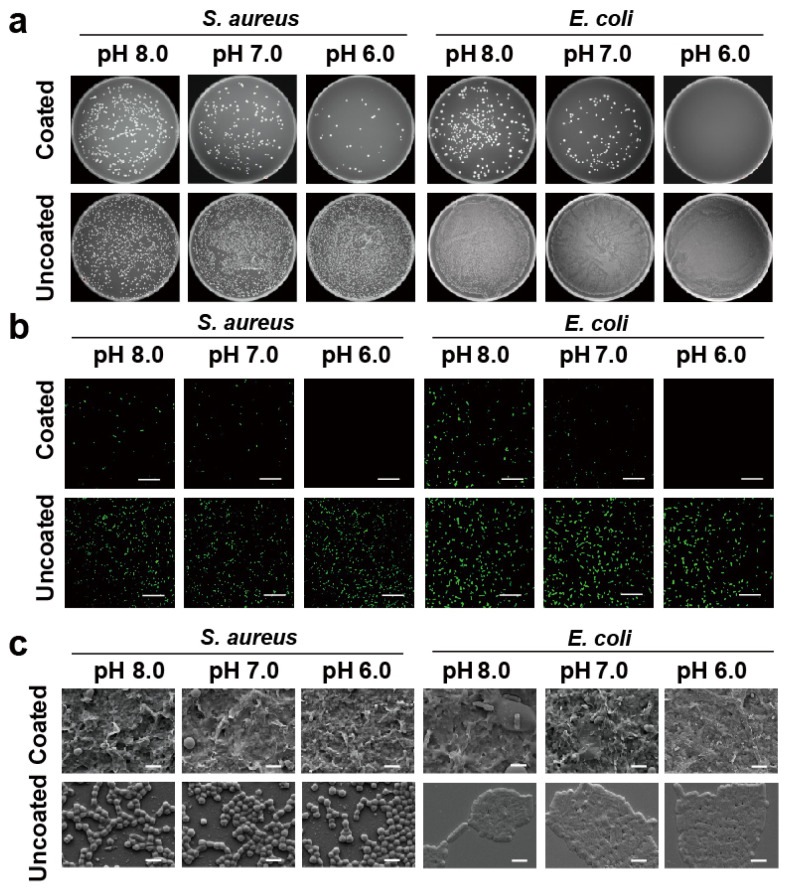
(**a**) Images of bacterial clones on the agar plates after treatment with (MMT-PPPB-CHA)_3_ coating for 2 h under different pH values (6.0, 7.0, and 8.0); (**b**) CLSM diagrams and (**c**) SEM images of bacterial adhesion on the (MMT-PPPB-CHA)_3_ coating at different pH values (6.0, 7.0 and 8.0) for 2 h. Scale bar: 20 μm and 1 μm.

**Figure 8 jfb-13-00024-f008:**
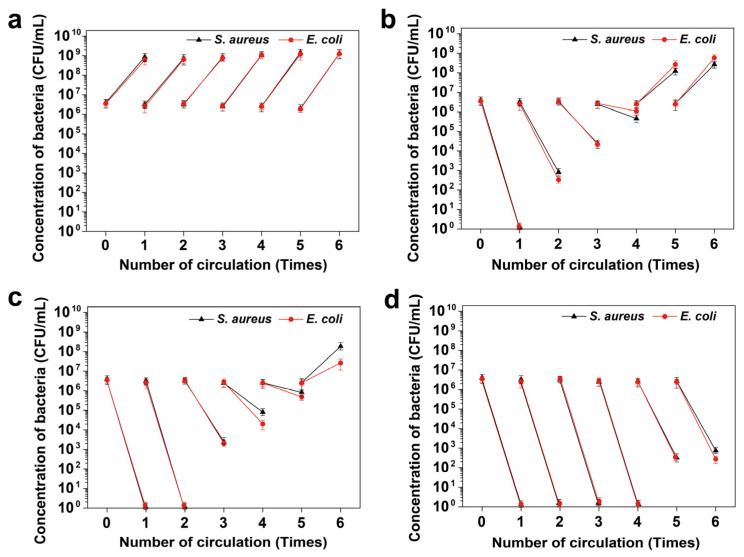
Bacteria cyclic killing effects of (**a**) Uncoated, (**b**) CHA coated, (**c**) (MMT-CHA)_3_ coated and (**d**) (MMT-PPPB-CHA)_3_ coated substrate against *S. aureus* and *E. coli*.

**Figure 9 jfb-13-00024-f009:**
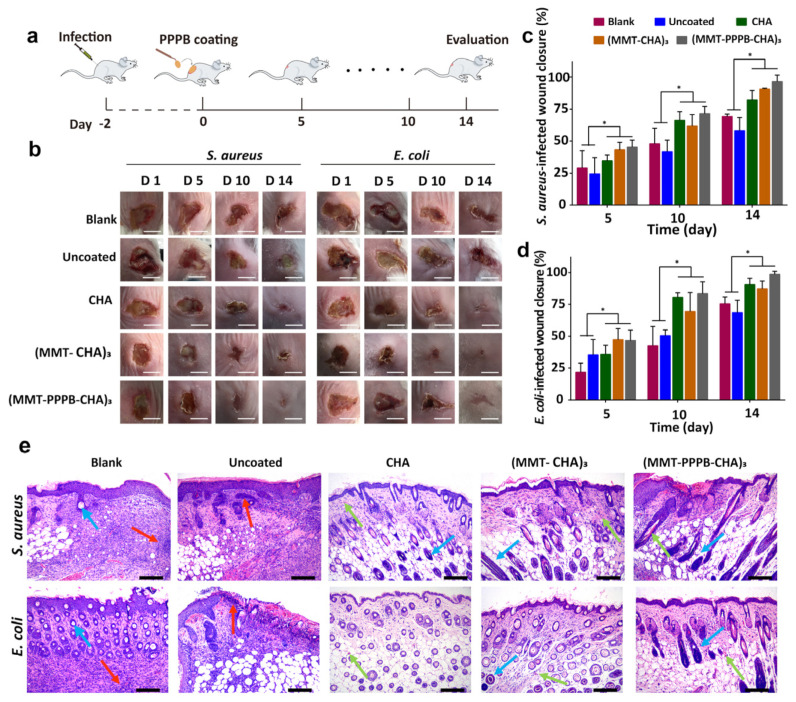
(**a**) Scheme of test of antibacterial activity and full-thickness skin defect healing in vivo; (**b**) Digital photographs of infected wound healing in vivo at days 1, 5, 10, and 14 after different treatments. Scale bar: 1 mm; (**c**) *S. aureus* and (**d**) *E. coli* infected wound closure rates during the 14 days in treatment; (**e**) H & E-staining results of healed skin tissues at determined times (blood vessels: green arrows, hair follicles: blue arrows, neutrophils: red arrows). Scale bar: 100 μm. * *p* < 0.05.

**Table 1 jfb-13-00024-t001:** The result analysis of L_9_(3^4^) orthogonal test.

Groups	Factors	CHA Release Rate (%)
pH	Glucose (mg/mL)	DTT (mg/mL)
a	6.0	0	0	56.0 ± 3.2
b	6.0	0.5	0.5	60.1 ± 2.0
c	6.0	1.0	1.0	70.1 ± 1.3
d	7.0	0	0.5	56.7 ± 1.7
e	7.0	0.5	1.0	62.0 ± 2.3
f	7.0	1.0	0	53.9 ± 1.1
g	8.0	0	1.0	43.3 ± 1.6
h	8.0	0.5	0	34.3 ± 1.8
i	8.0	1.0	0.5	50.7 ± 2.8
K1	186.2	156.0	144.2	
K2	172.6	156.4	167.8	
K3	128.3	174.7	175.4	
k1	62.1	52.0	48.1	
k2	57.5	52.1	55.9	
k3	42.8	58.2	58.5	
R	19.3	6.2	10.4

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
