# Peer review of "Multi-Stimulus Responsive Multilayer Coating for Treatment of Device-Associated Infections"

_jfb, 2022, doi:10.3390/jfb13010024_

Round 1

Reviewer 1 Report

Looking at stages in biofilm establishment, after adhesion, bacteria interact with each other to form microcolonies promoting bacterial aggregation. Large bacterial aggregates called as towers develop when polymeric matrix in biofilm progresses. PIA and eDNA expression in biofilm also contributes to the biofilm formation. Phenol-soluble modulins form characteristic water channels and are involved in bacterial dispersal. Does the antibacterial solution developed in this study have been effective any of the sub-stages of the biofilm establishment? Please give an explanation while discussing the results.

Line 66, statement ‘(e.g. weak acid environment…………’, I guess authors shall revise and quote as most of biofilm are generally acidic (pH < 5).

I find DTT stimulus concept in figure 1 weak; either clarify in figure legends r show in scheme more clearly?

In Figure 1. Schematic illustration of the multi-stimulus responsive multilayer antibacterial coating 106

(MMT-PPPB-CHA)n, please clearly quote or show with cartoons which is stimulus and which is response for the readers.

Materials and Method

Please provide the catalogue number (e.g. CHA, PEG,DTT, EDC, NHS, CPBA etc. and many others catalogue number missing), supplier country/region of all chemicals used. This is now mandatory for all studies to provide all details to avid interlaboratory variation of results in case researchers wish to replicate it to reduce the interlaboratory results variations.

Lin 254-256, cite https://doi.org/10.1371/journal.pone.0175428, to support the statement “….ImageJ was used to measure the change of wound area at different periods to calculate the wound healing rate”.

Similarly provide model number of all instruments and software versions used in the study.

Section 2.9, The absorbance of the sample was measured at 252 nm by an UV-VIS spectrophotometer, which component absorbs at this specific wavelength provide refs, similarly highlight for other molecules for the absorbance read out in the study?

What is purpose orthogonal test L9 (34) (three factors and three levels), provide more details and reference for the same?

Line 229- 233, how authors prevented the delamination and detachment of bacterial colonies from shear forces while intensive washing steps?

Which software/program was used for the 2.14 Statistical Analysis, provide details.

Provide ethical committee reports to carry out animal testing?

Results and discussion

Line 309, what are sedimentary layers?

Line 310, change thicker and looser to thick and loose

At what pH, glucose and DTT concentration based stimuli, responsiveness was tested? Overall what was optimal condition for the 3 stimuli tested for most efficacious colony inhibition, clearly mention in the results and discussion.

Figure 5. CHA release curves of (MMT-PPPB-CHA)n within 72 h in (a) different pH (5.5, 6.0, 7.0 and 361

8.0), (b) concentrations of glucose (0, 0.5, 1.0 mg/mL) and (c) DTT (0, 0.5, 1.0 mg/mL) solutions (n = 362

3), were these differences statistically significant?

Line 461-463, cite a latest report https://doi.org/10.1016/j.msec.2021.112592 with sentence ‘The skin tissues in the untreated group and the blank substrate group exhibited acute inflammatory responses and displayed very little sprouting angiogenesis on the 14 days’ to make the reference list up to date.

Conclusion

In real clinical scenario, can author comment on how toxic will be this formulation for the mammalian cells in case applied for the stents and prosthetic devices in vivo in Conclusion section?

Reviewer 2 Report

The authors report on the generation of a layer-by-layer coating with a multi-stimulus function for the release of an antibiotic molecule. The synthesized polymer is responsive to pH, glucose and DTT. The idea is interesting and worth publishing. The manuscript is well written, but there are some general issues that are sometimes confusing. Those should be modified and improved before accepting the manuscript for publication.

General issues:

  • The abstract contains too many abbreviations, which are also not explained (e.g. the antibiotic CHA, PPPB or MMT). The coating can for example just be mentioned as "coating" without any chemical details.
  • Layer-by-layer technology is usually abbreviated as LbL (with an uncapitalized b). Please revise throughout the manuscript. See e.g. https://doi.org/10.1021/acs.chemrev.6b00627
  • The "n" in (MMT-PPPB-CHA)n is a bit confusing. I would suggest to label the n with "3", as in the main manuscript only data with n=3 is used. Furthermore, "n=3" can easily be mistaken with the number of independent repeats. This especially in the Figure captions. The same for the suppl. Information, where sometimes the "n" is not given and it is not clear how many layers were used.
  • Provide more details on the materials in the introduction. For example, there is nothing mentioned why MMT was used and what it is.
  • Statistics: it is not clear what the error bars in all the graphs mean. Are these independent repeats. Please clarify and also describe in the M&M section.

Abstract:

L25: The abbreviation is not clear here. Please describe first what is meant with all these abbreviations. Also I would suggest to reduce the number of abbreviations in the abstract to a minimum.

L25: what means the "n" in (MMT-PPPB-CHA)n?

L26: LBL -> LbL

Introduction:

L85: "paradigm coating". This term implies too much of for this statement and LbL is a standardized coating technique.

L88: the LbL method could be described a bit more in detail, i.e. one sentence about the general concept. I think it is also worth to mention the pioneering work of Frank Caruso on the LbL technology (e.g. https://doi.org/10.1021/acs.chemrev.6b00627).

L92: What is the role of MMT? Why was this material chosen for the coating?

L95: Please mention what the abbreviations of PCA, PEG and BPBAC are.

Figure1: It is a bit confusing to show a layer of CHA, as it is a small molecule only and not a big molecule or mineral like the PPPB or MMT, respectively. It is also not clear, what the mechanism of the LbL method is. Please add for example some charge interactions. And maybe use for the CHA layer only single positive charges.

M&M:

L173: Please describe the zeta-potential measurement in more detail. I assume it was a surface-zeta potential measurement? For example, which buffer was used for the measurement?

Results:

L265: Please clearly describe one more time where the abbreviation PBBB is coming from. Same for BPBAC and PCA.

L296: In this section it is not clear how the LbL coating was done. Please describe also the ionic interaction between the different layers. As electrostatic interactions between the layers is the main reason for the LbL assembly, please describe in more detail the assembly based on the zeta-potential graph.

Also, what are the zeta-potentials of the surfaces used? Please describe why PEI was needed to be added to the substrates before adding the polymers.

L333: curve fitting: What do the numbers mean that the authors obtained from the fitting? I would remove the linear equation in the text, as it doesn't provide any information.

Figure5: For better comparison, please provide the graphs with same scale, ranging from 0 – 70%.

L379: The release of CHT shows first a burst release, which is probably excess CHT that is not tightly bound to the polymer layers. Therefore, the analysis of the release should be performed after incubation of the coating in a certain media (or water) before doing the antibacterial tests. This would probably provide a much better view on the responsiveness of the coating. At the moment, the It is obvious that bacteria are killed

L439: it is not clear to me, how the substrates were placed on the mice. Was it placed on the wound and then taken away again? Please describe more clearly.

Other comments:

  • Sometimes the word "bacterials" is used (e.g. in Figure 8 and in the Supp Info). Please change to "bacteria".

Round 2

Reviewer 1 Report

accept in present form

Author Response

There are no comments from the reviewer.

Reviewer 2 Report

Dear authors,

Most of the comments have been addressed accordingly. However, I found some minor issues, which should still be implemented.

L98: Please remove "According to Frank Caruso", as it implies that he was the only one who stated this information. Just write: "The LbL technology …"

The number of layers has now been added with a number. For better clarity, I would write the number as subscript.

Zeta-potential: please provide the experimental details in the Materials&Methods section. The experimental details are less important in the results section. Also I am not sure how pieces of the membrane are possible to measure using a particle analyzer. Those usually require nanoparticles homogenously dispersed in a solution, which I guess was difficult to achieve for a film that had to be first scraped off a substrate. Also, the scraping of the film results in two different exposed layers within the water. Proper zeta-potential measurements of surfaces require either "surface zeta-potential" measurements, or as alternative to zeta-potential, the QCM technique to show the adsorption of the alternating layers on top of each other. 

Figure 4: the figure caption reads a bit weird. Please change to e.g. "SEM micrographs of cross-sections of a) …". Also, is the substrate also visible on the SEM micrographs or is the "scraped-off" membrane shown? Please clarify.

Figure 5: The CHT release of the graph without glucose and without DTT, why are they not the same?

Figure 8 stills says "bacterials" in the graphs. Bacterials is not a correct English expression.

Answers to comments:

20) L379: The release of CHT shows first a burst release, which is probably excess CHT that is not tightly bound to the polymer layers. Therefore, the analysis of the release should be performed after incubation of the coating in a certain media (or water) before doing the antibacterial tests. This would probably provide a much better view on the responsiveness of the coating. At the moment, the It is obvious that bacteria are killed.

Response: In fact, that's what we do. Before use, the coating was soaked in pure water to remove excess CHA molecules. But the coating remains highly bactericidal, even after repeated use. According to Figure 8d, (MMT-PPPB-CHA)3 coating completely killed S. aureus and E. coli in four cycles. In the fifth and sixth cycles, low concentrations (103 CFU/mL) of S. aureus and E. coli appeared on the surface of the coating, respectively. It could be thus concluded that (MMT-PPPB-CHA)3 coating can maintain a long-term antibacterial effect.

Response: Please provide the information in the M&M section about the washing step. This is important in which experiments the washing step was performed and in which ones not.
